# Validity and Reliability of a Smartphone Application Versus 2D Software for Joint Range of Motion Measurement: A Cross-Sectional Validation Study

**DOI:** 10.3390/muscles4010008

**Published:** 2025-03-17

**Authors:** Inès Martins, Misha Gunot, Amândio Dias

**Affiliations:** 1Egas Moniz Center for Interdisciplinary Research (CiiEM), Egas Moniz School of Health and Science, 2829-511 Caparica, Portugal; 2Integrative Movement and Networking Systems Laboratory (INMOV-NET LAB), Egas Moniz Center for Interdisciplinary Research (CiiEM), Egas Moniz School of Health & Science, 2829-511 Caparica, Portugal; 3Sport Physical Activity and Health Research & Innovation Center (SPRINT), 2829-511 Rio Maior, Portugal

**Keywords:** Kinovea, PhysioMaster, mobile phone, reliability, passive range of motion, muscle mobility

## Abstract

The assessment of joint ranges is an essential component of the physical examination, enabling monitoring and follow-up. Additionally, it is a key component of muscle mobility assessment. Smartphone applications for range-of-motion assessments offer a cost-effective alternative to traditional measurement tools, reducing the need for expensive equipment while maintaining accuracy. Their portability and ease of use provide significant advantages in clinical settings, allowing for quicker assessments. This, in turn, can enhance patient care by enabling more frequent monitoring and timely interventions, ultimately improving treatment outcomes. This study evaluated the validity and reliability of the PhysioMaster application in measuring knee range of motion. Twenty-nine participants performed passive knee extensions, with data collected simultaneously through the application and video recordings for posterior angle calculations. The application demonstrated excellent validity, with intraclass correlation (ICC = between 0.729 and 0.814) and the Pearson correlation values ranging from r = 0.908 to 0.974. For inter-rater reliability, ICC was 0.898 and Pearson correlation coefficient r = 0.82. Additionally, the coefficient of variation was 5.18%, and the measurement error was 0.82°. The results showed that the PhysioMaster application is a valid and reliable tool for assessing passive knee extension in clinical environments, supporting efficient and accurate patient evaluations.

## 1. Introduction

Worldwide, 1.71 billion people suffer from musculoskeletal dysfunctions, and these pathologies lead to functional limitations that reduce quality of life [1,2]. This type of dysfunction causes long-term disability with great expenditures on health and social resources [3]. They are characterized by pain and temporary limitations in mobility [4] and are among major risk factors for several diseases, such as osteoporosis and sarcopenia [3]. Physiotherapists as well as practitioners in the field of sports science/strength and conditioning play a crucial role in the rehabilitation of people suffering from musculoskeletal dysfunctions, improving muscle strength, flexibility, joint range of motion, and quality of movement in order to restore the individual’s functionality [5]. The assessment of joint ranges is an essential component of physical examination performed by physiotherapists allowing for more reliable follow-up. These measures are generally important in the development of the professional’s objectives in relation to the individual and the treatment for the rehabilitation process [6].

The universal goniometer (UG) is the most common method used by physiotherapists to perform these assessments, as it is inexpensive, portable, and quick to use [6]. However, in addition to the UG, there are several methods for measuring the range of motion of a joint. One of these methods is the three-dimensional analysis of movements (3D) [7]. Moreover, 3D kinematic analysis requires substantially more expensive equipment that is difficult to access in clinical practice. On the other hand, 2D kinematic analysis is more accessible and easier to use to measure angles of movement during dynamic movement. Additionally, this type of equipment presents a validity that can be compared to 3D systems [8].

Among the various solutions available, the Kinovea software (version 2023.1.2) is a free, intuitive, and widely used tool for 2D kinematic analysis. It enables kinematic parameters such as joint distances or angles to be determined from previously collected images/videos [9,10]. Several studies have already used this software to validate the measurement of joint angles in some movements, such as jumping and flexibility tests of the hamstring muscles [9,11,12]. In addition, other studies have used Kinovea as a reference measure for validation [13]. However, despite its ease of use, its application in clinical practice is limited by the need for post-use digitization on a computer. Therefore, the search for valid tools that are simpler to use and provide feedback in real time is an emerging requirement in clinical practice.

The great technological growth we are witnessing is helping to improve and optimize assessment and monitoring processes in physiotherapy [14]. Several studies have shown that there are valid and reliable applications for measuring knee range of motion, as in other body segments [14,15,16,17]. However, the reliability is not consistent, since it ranges from moderate in the hip joint [14], to very strong in the hip [16] and knee joints [15]. Therefore, more studies are required on this topic.

Given that 75% of the world’s population over the age of ten uses a cell phone with internet access [18], we can assume that a large number of physiotherapists own a smartphone, making it a universally accessible, non-invasive, and easy-to-use portable tool to facilitate clinical examinations [18,19,20].

Although there are several applications on the market, a recent systematic review [21] reported that most of the studies that validated these smartphone applications used the UG as the validation method. Given the limitations of the UG, it is important that applications are validated using kinematics. Another limitation of most currently validated apps is that they are either fee-based or limited to a single operating system. For example, the Dr Goniometer app is only available on the IOS operating system, while the Hip ROM Tester is only available on the Android operating system [22,23]. However, practitioners should be aware of smartphone placements and measurements procedures, since measurements from smartphones may be erroneous if the smartphone is placed in the wrong orientation [24].

Considering the previous aspects, the purpose of this study is to assess the validity and reliability of a smartphone application, compared to 2D kinematic analysis, for measuring passive knee joint range. The choice for passive movement was to allow for articular movement without the interference of muscle contraction—more specifically, the PhysioMaster application [25], which is freely accessible and compatible with both IOS and Android systems, making it accessible to a wide range of populations, regardless of their device. Based on the previous literature, our hypothesis is that the application has good validity and reliable measurement of joint amplitude [15,26,27].

## 2. Results

Table 1 presents participants’ characteristics.

Table 2 presents a descriptive analysis of the subject’s evaluation, for both evaluators and the three observers.

Table 3 shows the Intraclass Correlation Coefficient (ICC) values between the three observers and both evaluators (AV1 and AV2), which range from 0.729 to 0.814, while Pearson’s correlation coefficient values vary from 0.908 to 0.974.

Based on the statistical results, an excellent correlation between the data was found for AV1, both for ICC and Pearson’s correlation, with statistically significant differences. This pattern of high agreement was also observed between AV2 and the three observers, where the ICC was between 0.729 and 0.746 and the Pearson correlation coefficient values ranged from 0.908 to 0.926, which are considered very strong.

Table 4 shows the reliability between the three observers when analyzing data collection by AV1. The results indicate excellent concordance, with ICC values between 0.995 and 0.998, and Pearson’s correlation coefficient values between 0.990 and 0.995. In addition, the coefficients of variation are low between observers, as are the measurement errors, thus reinforcing the accuracy of the results.

Table 5 shows the inter-observer reliability for the data collected by AV2. As reported for AV1, excellent reliability was also observed, with intraclass correlation values between 0.995 and 0.996 and Pearson’s correlation coefficient values between 0.991 and 0.993, with significant differences. Lower coefficients of variation and measurement errors also suggest greater precision in the results.

Finally, Table 6 shows the reliability of the data collected with the smartphone between the two evaluators. The results show excellent reliability, with low variation, expressed by the coefficient of variation and standard error of measurements between the evaluators.

Additionally, Bland–Altman plots were also made (Figure 1), in order to assess any potential bias. The plots show a high level of concordance between measurements both for evaluators (Physiomaster) and observers (Kinovea).

With regard to the data analysis and results, no outliers or anomalies were present.

## 3. Discussion

According to the results, the smartphone application can be considered valid when compared to the 2D kinematic analysis. In fact, the data in Table 3 reveal that the ICC shows excellent reliability (ICC = 0.802 to 0.814) and a very strong correlation (r = 0.970 to 0.974) between the first evaluator (AV1) and the three observers, in other words, between the data obtained with the PhysioMaster application and the Kinovea data. In the same table, we can also observe good reliability (ICC = 0.729 to 0.746), and, regarding Pearson’s correlation, there is a very strong correlation (r = 0.908 to 0.926) between AV2 and the three observers. In addition, the application proved to be reliable.

In fact, there was excellent reliability (ICC = 0.898) and a strong correlation between AV1 and AV2 (r = 0.82) (Table 6). However, the results also showed considerable inter-rater measurement variability, due to an elevated coefficient of variation (CV = 5.18%), which is a value less than ideal [28]. In addition to the high coefficient of variation, the SEM is 3.0282°. However, according to Chaudhary et al. [29], the minimum measurable difference is 5°, making this value not clinically relevant. A possible hypothesis for this variability may be related to biological variation between evaluators [28], since both had prior knowledge and experience with the smartphone application. The evaluators in the present study were physical therapy students with prior training in performing the test movement, but maybe the lack of experience could have played a role in the variability that was present. Another possible explanation is the lack of prior training on the testing protocol that was used. Despite conducting pilot tests before the data collection, it could be possible that, since there is no standard training and data protocol for this kind of testing, the setup procedure used in the present study was not the most ideal.

The study by Castle et al. [30] assessed active knee flexion and extension by comparing the data obtained with the GU and the Dr. Goniometer smartphone app. The results showed good validity and high reliability, with an inter-rater ICC ranging from 0.842 to 0.947 for extension and 0.981–0.995 for flexion. Like previous authors, Mehta et al. [31] assessed the reliability and validity of the iPhone Goniometer app (i-Goni) by comparing it with the GU. The results showed excellent reliability for active flexion (ICC = 0.93) and good reliability for active extension (ICC = 0.81). Regarding validity, the highest correlation was observed for assessing knee flexion (r = 0.92; *p* < 0.01), while the correlation between the i-Goni and the GU for assessing knee extension was moderate (r = 0.68, *p* < 0.01). Hancock et al. [32] evaluated the inter-rater reliability of visual estimation (ICC = 0.991), the GU with a short arm (ICC = 0.991) and long arm (ICC = 0.996), the Dr Goniometer pro app (ICC = 0.994), and the Halo digital goniometer (ICC = 0.999) for active flexion and extension. These authors reported excellent reliability. Moreover, from a methodological point of view, they used the same anatomical reference points as the present study.

Two studies had very similar objectives and methodologies to the present study [15,26]. However, it is not possible to make a direct comparison between the results due to incoherencies in the methodology and results/statistical analysis. In fact, in the studies by Derhon et al. [26] and Dos Santos et al. [15], the methodological description explains that the movement performed was knee flexion, but the figures representing this assessment show a knee extension movement, in both studies. They also used Bland–Altman’s statistical method to measure inter- and intra-rater reliability, which is generally used to compare two instruments [26,33].

Although the above-mentioned studies gave good results in terms of reliability and validity of the applications evaluated, some studies suggest that the use of the GU on lower limbs may contain errors of 5 to 10 degrees, even when the evaluator is experienced [34,35,36]. Therefore, comparison/validation with the GU may be compromised due to this variation. Additionally, the study by Ockendon and Gilbert [37], which assessed static knee flexion movement, revealed that the iPhone’s digital goniometer has greater reliability and correlation when compared to the Lafayette goniometer. In addition, the study by García-Rubio et al. [6] assessing dorsal flexion of the ankle and comparing the measurements obtained with the Kinovea and the GU showed that the results were more reliable and that the margin of error was lower when the 2D software was used compared to the goniometer, thus justifying the methodology of this study and the reliability of the results.

The results in Table 4 and Table 5 show excellent inter-observer reliability (ICC = 0.995 to 0.998) with a very low standard error of measurement (SEM = 0.4896° to 0.7256°). This means that there are very few differences in degrees between the three observers in the 2D software, helping to confirm Kinovea as a tool with a smaller margin of error, as previously mentioned [6].

Regarding the technique for fixing the smartphone to the participant, there are other studies that have used a similar method to this study but with different articulations.

Charlton et al. [14] used adhesive tape to attach the cell phone to the participant’s thigh and measure the passive amplitudes of hip abduction and adduction. The results reported for inter-rater reliability (abduction: ICC = 0.68; adduction: ICC = 0.68) were considered good by the authors. In another study [38], which also assessed the reliability of an app with the hip joint, a customized cell phone case was used to fix the device to the tibia and measure passive internal rotation of the hip. The results found for inter-rater and intra-rater reliability varied from moderate to good (ICC = 0.72 to 0.80).

In this study, knee extension was measured passively. In the studies that used the GU as a reference to validate the smartphone application, it seems that the movement used (passive or active) had no influence on the results. In this way, studies using passive or active movements show good consistency in the results between assessment instruments [14,29,30,31,32,38]. Even though in the present study the application was compared with 2D software, we can recognize that the method of assessing movement had no influence on our results.

The aim of this study was to validate a smartphone application in order to confirm whether it can be used in clinical practice. The application mimics a similar role to the goniometer in assessing passive movements. For this reason, passive movement was chosen for the measurements [39].

Physical therapy and rehabilitation require instruments that are portable and easy to use for assessing measurements in treatment protocols [40]. Applications from smartphones are fast and portable and allow practitioners to obtain measurements without restrictions. The knee movement and range of motion (ROM) are key parameters for evaluating knee function, and, particularly, passive movements are an indicator of the patient’s functional status [41]. So, it appears that the use of smartphones for ROM measurements can assist physical therapists with their clinical evaluations of patients.

For example, with elderly patients, assessment with a smartphone could help assess daily movement limitations, such as walking up a flight of stairs. For patients with musculoskeletal conditions, this type of assessment could help monitor daily changes and movements in a more objective manner.

Some limitations emerged during this study. Despite being available on IOS and Android, the application is not compatible with some older smartphone models. This may be due to specific hardware requirements, minimum operating system versions, or regional restrictions. Practitioners considering using this smartphone app should do so by using a newer smartphone, with an updated operating system, to ensure that there is no software incompatibility. Furthermore, even though fixing the cell phone to the participant’s lower limb reduces the risk of bias and has shown good results in other studies, this technique does not adequately represent clinical practice and is not compatible for all joints.

The results of the present study can only be related to healthy young adults. Future studies will be necessary to validate the use of this application in a more diverse population pertaining to age in order to extrapolate the results and thus be more representative of the clinical environment. Additionally, it would be of interest to test the validity of this smartphone application with participants that present different musculoskeletal conditions and also in other body joints. It would also be appealing to carry out future studies to assess the reliability of the application by comparing experienced and non-experienced evaluators.

## 4. Materials and Methods

### 4.1. Study Design

This is a cross-sectional, observational, validation study. Each participant visited the laboratory for one day only, with an estimated total duration of 30 min.

### 4.2. Study Settings

This study was performed in a university setting by two students in their final year of physiotherapy at the Egas Moniz School of Health. The data were collected at the Egas Moniz physiotherapy clinic under the supervision of a university professor. The study was approved by the ethics committee under number 1267.

### 4.3. Participants

Study participants were included according to the following criteria: age between 18 and 75 and the ability to understand and communicate with the evaluators. All participants with cognitive dysfunctions, hypersensitivity to pain, infectious diseases, fractures or other contraindications to movement were excluded [42].

Participants were recruited by social media and flyers at the university campus. Patients from the physiotherapy university clinic and students from the different courses were also invited to take part.

### 4.4. Sample Size Calculation

With an alpha level of 0.05, an approximate size of 27 participants was estimated (with 4 measurements for each) to provide the present study with 95% power to detect an ICC of 0.85 in relation to the null hypothesis of 0.65 [43,44].

The initial sample was composed of thirty-three participants. After data collection, four participants were excluded due to measurement errors. The final study sample consisted of twenty-nine participants.

### 4.5. Data Collection and Organization

Prior to any assessment, subjects provided informed consent, all procedures were explained, and all questions were answered. To carry out this study, we adapted the methodology previously used [26].

The evaluators were familiarized with the PhysioMaster application for 6 months prior to data collection and with the Kinovea software throughout the four years of their physiotherapy degree [25].

The smartphone used for the joint range measurements was an iPhone 12, IOS version 16.3.1. The tablet used to record the videos was iPad mini (5th generation) and iPadOS version 16.4.1(a). The PhysioMaster application, version 2.1.7, was used. Moreover, for data collection, a physiotherapy table and adhesive tape for attaching the cell phone to the skin were used.

The tablet was positioned on a tripod 2.4 m from the table and placed 25 cm from the floor parallel to the table, in order to film the movement of the lower limb in the sagittal plane [13]. A recording was made for each assessment, which started when the patient was in the correct position.

The assessment was always carried out on the right lower limb to make all the assessment procedures and methods uniform. The use of the same lower limb for assessment was also to reduce variability regarding natural differences between limbs, as well as dominance, which could influence ROM. Before starting the assessment, the participants were asked to position themselves in a supine position near the edge of the table with their thighs resting on the surface and the edge of the table touching the popliteal fossa, leaving their knees bent at around 90°. Marks were then placed on the following anatomical points: greater trochanter, femoral lateral condyle, and lateral malleolus. The marks used were circular adhesive surfaces 1.5 cm in diameter, identical to those used in other studies [13]. These marks were placed while the participant was already lying on the couch to reduce the risk of sliding on the skin. For better positioning of the marks, participants were asked to wear shorts so that the anatomical points were more visible, and the adhesive marks were placed directly on the skin. In front of the participant, perpendicular to the edge of the table and at the same height, a horizontal bar was placed about 40 cm away in order to reduce the influence of the participant’s subjective sensation of hamstring stretching [26,45]. After placing the adhesive marks, the reference for placing the upper part of the smartphone on the lower limb was marked, five centimeters below the tuberosity of the tibia. This reference was marked on the participant’s skin. The smartphone was taped to the lower limb to ensure that there were no accessory movements that could give false measurements. When all the previous steps were completed, the evaluator pressed the “START” button on the app to calibrate, considering 90° of knee flexion as the starting position (0°). Then, the evaluator supported the lower limb so that the markers were not covered, performed passive knee extension until it touched the horizontal bar, pressed the “STOP” button, and recorded the measurement. The previous methodological procedures are exemplified in Figure 2.

### 4.6. Data Analysis

The two evaluators (AV1 and AV2) carried out two evaluations alternately, and the order in which they were conducted was determined randomly, resulting in a total of four evaluations per subject. Data collection was blinded between evaluators. At the end of each assessment, the data were registered. Thus, four evaluations were obtained using the PhysioMaster application for each subject in the sample. These values corresponded to the degrees of joint range available during passive knee extension. Additionally, four videos of each evaluation were also made. The videos were used, posteriorly to data collection, by three independent observers (OB1, OB2, and OB3) to perform a kinematic analysis, and angle measurements were obtained for each evaluation. The data collected from Kinovea by the observers were considered the reference for establishing the validity and reliability.

In order to preserve their anonymity, each participant was given a specific code on a whiteboard next to the table while recording the videos for analysis with Kinovea. The same code was used by both evaluators while using the PhysioMaster application.

### 4.7. Statistical Analysis

The Shapiro–Wilk test was used to define the distribution of the data to determine the normality and equality of the variances of the data, respectively.

Inter-rater reliability was determined by calculating the intra-class correlation coefficient (ICC), the coefficient of variation (CV%), the standard error of measurement (SEM°), and Pearson’s correlation (r).

Inter-observer reliability was estimated by comparing observer 1 (OB1) vs. observer 2 (OB2), OB1 vs. OB3, and OB2 vs. OB3, for each of the assessments performed. Concurrent validity between assessment instruments was determined by calculating the intra-class correlation coefficient (ICC) and Pearson’s correlation (r).

The ICC, with a 95% confidence interval, was interpreted as follows: 0.4 = poor reliability; 0.4–0.75 = moderate-to-good reliability; and >0.75 = excellent reliability [28] Pearson’s correlation was interpreted as follows: 0.1–0.39 = weak correlation; 0.40–0.69 = moderate correlation; 0.70 to 0.89 = strong correlation; 0.9–1.0 = very strong correlation [46].

The correlation between the application and Kinovea data was also assessed using Bland–Altman graphs [33] in order to check for bias.

All statistical tests were calculated using Jamovi software, version 2.3.28 [47].

## 5. Conclusions

This study investigated the use of a smartphone application to assess passive knee extension in healthy individuals. The results reveal high validity and reliability in measurements with the current setup. Despite considerable inter-rater variations, the application demonstrates good reliability and validity when compared to 2D software, confirming its usefulness. For this reason, the PhysioMaster application can serve as a useful tool in a clinical environment for assessing passive knee extension.

The results of the present study could have practical implications in several areas for practitioners. It can be used to assess patients recovering from knee injuries and/or surgical procedures, as well as injury prevention, since range of motion could limit athletes’ movement for performance or could even be used in a remote healthcare setting, where patients with guided instructions by a therapist could perform self-evaluation. Future studies should address the validity of smartphone application in other populations.

## Figures and Tables

**Figure 1 muscles-04-00008-f001:**
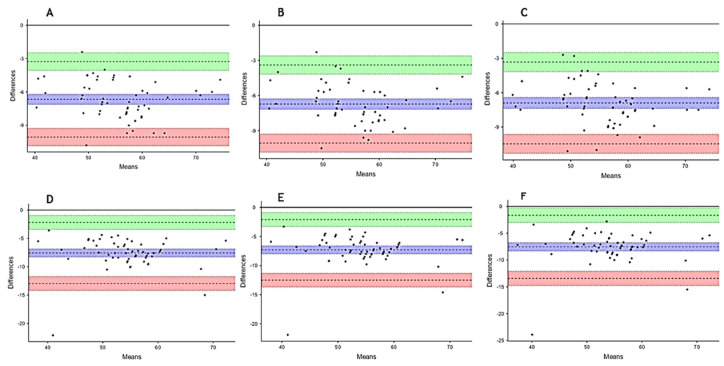
Bland–Altman plots. (**A**) OB1 vs. AV1; (**B**) OB2 vs. AV1; (**C**) OB3 vs. AV1; (**D**) OB1 vs. AV2; (**E**) OB2 vs. AV2; (**F**) OB3 vs. AV2.

**Figure 2 muscles-04-00008-f002:**
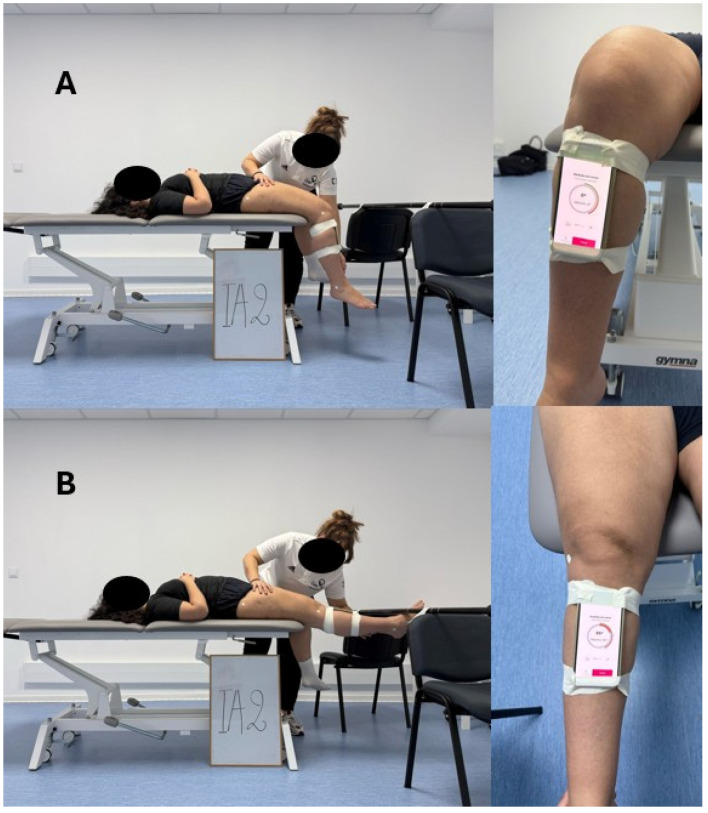
Setup for data collection. (**A**) Initial position; (**B**) final position.

**Table 1 muscles-04-00008-t001:** Subject’s characteristics.

Participants (n = 29)	
Female	22
Male	7
Age (years)	22 ± 2.9
Height (m)	1.68 ± 0.07
Weight (kg)	64.3 ± 11

**Table 2 muscles-04-00008-t002:** Data collection from the smartphone app for both evaluators (AV1 and AV2) and from the observers (OB1, OB2, OB3) on data collected from both evaluators.

	Evaluators		Kinovea					
	AV1	AV2	OB1 (AV1)	OB2 (AV1)	OB3 (AV2)	OB1 (AV2)	OB2 (AV2)	OB3 (AV2)
Mean ± Standard deviation	59 ± 7.47	57.9 ± 6.98	52.3 ± 7.14	52.3 ± 7.12	52.1 ± 7.09	50.4 ± 7.47	50.6 ± 6.84	50.4 ± 7.03

Data presented in degrees.

**Table 3 muscles-04-00008-t003:** Concurrent validity between data from evaluator 1 (AV1), evaluator 2 (AV2), and the three observers.

	Evaluator 1			Evaluator 2		
	ICC	95% CI	r	ICC	95% CI	r
OB1	0.814	−0.0556; 0.957	0.973 *	0.729	−0.114; 0.929	0.921 *
OB2	0.812	−0.528; 0.956	0.974 *	0.746	−0.114; 0.934	0.926 *
OB3	0.802	−0.583; 0.953	0.970 *	0.731	−0.134; 0.928	0.908 *

* *p* < 0.001; ICC: Intraclass Correlation Coefficient; 95% CI: Interval of confidence; r: Pearson correlation coefficient.

**Table 4 muscles-04-00008-t004:** Inter-observer reliability for data from evaluator 1.

	ICC	95% CI	CV (%)	SEM (°)	r
OB1 vs. OB2	0.998	0.996; 0.999	0.94	0.4896	0.995 *
OB1 vs. OB3	0.995	0.991; 0.997	1.39	0.7256	0.990 *
OB2 vs. OB3	0.995	0.992; 0.997	1.28	0.6694	0.991 *

* *p* < 0.001; ICC: Intraclass Correlation Coefficient; 95% CI: Interval of confidence; CV (%): coefficient of variation; SEM (°): standard error of measurement; r: Pearson correlation coefficient.

**Table 5 muscles-04-00008-t005:** Inter-observer reliability for data from evaluator 2.

	ICC	95% CI	CV (%)	SEM (°)	r
OB1 vs. OB2	0.996	0.993; 0.998	1.15	0.5793	0.993 *
OB1 vs. OB3	0.995	0.992; 0.997	1.31	0.662	0.991 *
OB2 vs. OB3	0.995	0.991; 0.997	1.34	0.6774	0.991 *

* *p* < 0.001; ICC: Intraclass Correlation Coefficient; 95% CI: Interval of confidence; CV (%): coefficient of variation; SEM (°): standard error of measurement; r: Pearson correlation coefficient.

**Table 6 muscles-04-00008-t006:** Reliability between evaluators.

	ICC	95% CI	CV (%)	SEM (°)	r
AV1 vs. AV2	0.898	0.882; 0.941	5.18	3.0282	0.82 *

* *p* < 0.001; ICC: Intraclass Correlation Coefficient; 95% CI: Interval of confidence; CV (%): coefficient of variation; SEM (°): standard error of measurement; r: Pearson correlation coefficient.

## Data Availability

The data presented in this study are available on request from the corresponding author.

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
