# Peer review of "Validity and Reliability of a Smartphone Application Versus 2D Software for Joint Range of Motion Measurement: A Cross-Sectional Validation Study"

_muscles, 2025, doi:10.3390/muscles4010008_

Round 1
Reviewer 1 Report (Previous Reviewer 3)
Comments and Suggestions for Authors Dear Authors: Thank you for submitting your revised manuscript. The paper has been substantially improved according to the reviewers' comments. However, I have a few minor suggestions that could further enhance the quality of your manuscript:- Title: While your current title is comprehensive, consider making it more concise while retaining the key information. For example: "Validation of a smartphone application versus 2D software for joint range of motion measurement: A cross-sectional study"
- Citations:
- The introduction would benefit from additional supporting references, particularly in the early statements about musculoskeletal dysfunctions and their impact
- Consider increasing citation density in sections discussing methodology and clinical applications
- Include recent (past 2-3 years) references to strengthen the contemporary relevance of your research
- Study Significance: In the abstract, consider strengthening the emphasis on clinical relevance and practical implications by:
- Adding a clear statement about the potential cost-effectiveness of smartphone applications
- Highlighting the practical advantages in clinical settings
- Explicitly stating the potential impact on patient care and clinical workflow
Author Response
Comment 1: Dear Authors: Thank you for submitting your revised manuscript. The paper has been substantially improved according to the reviewers' comments. However, I have a few minor suggestions that could further enhance the quality of your manuscript:
Response 1: Thank you for your feedback.
Comment 2: Title: While your current title is comprehensive, consider making it more concise while retaining the key information. For example: "Validation of a smartphone application versus 2D software for joint range of motion measurement: A cross-sectional study"
Response 2: Thank you for your comment. We have updated the title, to provide a more concise information on the addressed topic.
Comment 3: Citations:
The introduction would benefit from additional supporting references, particularly in the early statements about musculoskeletal dysfunctions and their impact
Consider increasing citation density in sections discussing methodology and clinical applications
Include recent (past 2-3 years) references to strengthen the contemporary relevance of your research
Response 3: Thank you for your comment. In the beginning of the introduction section, we have added more supporting information and references regarding the effects that musculoskeletal dysfunction could have.
Additionally, we have increased the citation density with regards to clinical application (advantage of these type of measurements in knee ROM assessments) with recent references, to help support our study.
Comment 4: Study Significance: In the abstract, consider strengthening the emphasis on clinical relevance and practical implications by:
Adding a clear statement about the potential cost-effectiveness of smartphone applications
Highlighting the practical advantages in clinical settings
Explicitly stating the potential impact on patient care and clinical workflow
Response 4: Thank you for comment. We have added the statements in early part of the abstract, prior to the objective of the study.
Reviewer 2 Report (Previous Reviewer 1)
Comments and Suggestions for Authors
Based on the review of the manuscript, I would recommend the following points to improve its overall clarity and structure:
Figures:
Figures 1 and 2 are of poor quality and need improvement to enhance readability and understanding. Replacing them with higher-resolution images or clearer graphical representations would be beneficial. Although in their comments they say they have been improved, these improvements are not substantially noticeable.
Congratulations on responding to all the comments made by previous reviewers and improving on all the comments made by these reviewers.
Author Response
Comment 1: Figures 1 and 2 are of poor quality and need improvement to enhance readability and understanding. Replacing them with higher-resolution images or clearer graphical representations would be beneficial. Although in their comments they say they have been improved, these improvements are not substantially noticeable.
Congratulations on responding to all the comments made by previous reviewers and improving on all the comments made by these reviewers.
Response 1: Thank you for your comment. We have added new figures, with increased contrast.
Reviewer 3 Report (New Reviewer)
Comments and Suggestions for Authors
Thank you for this manuscript that evaluates the validity and reliability of a smartphone app (PhysioMaster) for measuring passive knee extension range of motion in healthy individuals and comparing it to 2D kinematic analysis using Kinovea software. The study concludes that the application is both valid and reliable, suggesting its potential utility in clinical settings. It was interesting for me to read, but I have some suggestioons
Title: is clear and concise, reflecting the study's focus
Abstract: provides a good overview of the study's purpose, methods, and key findings, including ICC and Pearson correlation values.
Introduction: provides a solid background on the importance of joint range of motion assessment in physiotherapy and the limitations of existing tools like the universal goniometer (UG). The introduction could better justify why the study focuses on passive knee extension specifically, as opposed to other joint movements or active range of motion. Please briefly mention the limitations of smartphone apps (e.g., hardware compatibility issues) to provide a more balanced view.
Materials and Methods: well-described, including study design, participant recruitment, data collection procedures, and statistical analysis. However, the section on data collection could clarify why the right lower limb was chosen exclusively and whether this could introduce bias.
Results: are presented clearly, with tables and statistical values (ICC, Pearson correlation) that support the study's conclusions. The results could briefly mention any outliers or anomalies in the data, if present.
Discussion: compares the study's findings to previous research and highlights the potential clinical utility of the PhysioMaster app. The discussion could also consider the generalizability of the findings to other joints or populations, such as individuals with musculoskeletal conditions.
Conclusion: summarizes the study's findings and highlights the potential clinical applications of the app. The conclusion could also suggest directions for future research, such as validating the app in diverse populations or for other joints.
General statement: The manuscript is well-written and provides valuable insights into the validity and reliability of the PhysioMaster app. However, it would benefit from some minor flaws as mentioned above Addressing these points would strengthen the manuscript and make it suitable for publication in a your journal.
Author Response
Comment 1: Thank you for this manuscript that evaluates the validity and reliability of a smartphone app (PhysioMaster) for measuring passive knee extension range of motion in healthy individuals and comparing it to 2D kinematic analysis using Kinovea software. The study concludes that the application is both valid and reliable, suggesting its potential utility in clinical settings. It was interesting for me to read, but I have some suggestions
Title: is clear and concise, reflecting the study's focus
Response: Thank you for your comment.
Comment 2: Abstract: provides a good overview of the study's purpose, methods, and key findings, including ICC and Pearson correlation values.
Response 2: Thank you for your comment.
Comment 3: Introduction: provides a solid background on the importance of joint range of motion assessment in physiotherapy and the limitations of existing tools like the universal goniometer (UG). The introduction could better justify why the study focuses on passive knee extension specifically, as opposed to other joint movements or active range of motion. Please briefly mention the limitations of smartphone apps (e.g., hardware compatibility issues) to provide a more balanced view.
Response 3: Thank you for your comment. We have added the reasoning for the passive movement in the purpose of the study, as well addressed smartphone measurements limitations and incorrect measurements procedures.
Comment 4: Materials and Methods: well-described, including study design, participant recruitment, data collection procedures, and statistical analysis. However, the section on data collection could clarify why the right lower limb was chosen exclusively and whether this could introduce bias.
Response 4: Thank you for your comment. As stated in the manuscript, the measurements were all on the lower right limb; to make all procedures and methods uniform across participants. We added additional clarification in the manuscript: To reduce variability, as assessing the same limb eliminates variations associated with possible natural differences between the right and left limbs. As dominance can influence muscle amplitude and strength, systematically choosing the same side (right) ensures that this variable is constant in all participants
Comment 5: Results: are presented clearly, with tables and statistical values (ICC, Pearson correlation) that support the study's conclusions. The results could briefly mention any outliers or anomalies in the data, if present.
Response 5: Thank you for your comment. No outliers were present in the results. We have added this information in the end of the results section.
Comment 6: Discussion: compares the study's findings to previous research and highlights the potential clinical utility of the PhysioMaster app. The discussion could also consider the generalizability of the findings to other joints or populations, such as individuals with musculoskeletal conditions.
Response 6: Thank you for your comment. Given that we only validated the smartphone application with young healthy adults, we didn’t want to generalize to other joints or populations, since procedures as well as patients’ mobility issues (in people with musculoskeletal conditions) could render the assessment invalid. However, we addressed that topic on the final paragraph of the discussion section.
Response 7: Conclusion: summarizes the study's findings and highlights the potential clinical applications of the app. The conclusion could also suggest directions for future research, such as validating the app in diverse populations or for other joints.
Comment 7: Thank you for your comment. Suggestions for future studies were already discussed in the final part of the discussion section, but that reference in the conclusion section, as recommended.
Comment 8: General statement: The manuscript is well-written and provides valuable insights into the validity and reliability of the PhysioMaster app. However, it would benefit from some minor flaws as mentioned above Addressing these points would strengthen the manuscript and make it suitable for publication in your journal.
Response 8: Thank you for your comment. We hope that the answers we provided as well and the changes made to the manuscript meet your expectations and are to your satisfaction.
Reviewer 4 Report (New Reviewer)
Comments and Suggestions for Authors
Interesting article on an application to mobility ranges.
Introduction
Please do not forget physical activity professionals who work more deeply than physiotherapists in rehabilitation and improvement of conditional capacity.
Some validated applications of published studies should be cited (for example, MyJump in the MyLab application) and the level of reliability should be rated.
The Materials and Methods section should appear prior to the results.
Methods
The age range of the participants was excessively wide. Several classifications are required for comparisons.
The minimum quality that the device must have to use for the application should be indicated.
The image that appears does not allow us to see the details; therefore, it is recommended to use one in the foreground.
There is no image in the software, and it would be ideal to include an illustrative image.
Discussion
Data compared with other studies are indicated, but there are no classifications by age or sex, which should appear in the limitations as it does not clarify how to apply the findings.
Author Response
Comment 1: Interesting article on an application to mobility ranges.
Response 1: Thank you for your kind appreciation of our manuscript.
Comment 2: Introduction. Please do not forget physical activity professionals who work more deeply than physiotherapists in rehabilitation and improvement of conditional capacity.
Response 2: Thank you for your comment. You are absolutely right. We have added the reference to those professionals in the introduction section.
Comment 3: Some validated applications of published studies should be cited (for example, MyJump in the MyLab application) and the level of reliability should be rated.
Response 3: Thank you for your comment. In the Introduction section we have already cited several published studies regarding range of motion assessment with smartphones (references #14 to #17). We have added the reliability results of those studies, to better support the requirement of our study.
Comment 4: The Materials and Methods section should appear prior to the results.
Response 4: Thank you for your comment. The structure of the manuscript is in accordance with the Muscles journal template file, available at
https://www.mdpi.com/journal/muscles/instructions
Comment 5: Methods. The age range of the participants was excessively wide. Several classifications are required for comparisons.
Response 5: Thank you for your comment. As reported in table 1, mean age was 22 ± 2.9, with all participants having an age between 18 and 24. Since the study was to be conducted on heathy young adults, we feel that the age requirements are well adjusted to the study.
Comment 6: The minimum quality that the device must have to use for the application should be indicated.
Response 6: Thank you for your comment. The minimum quality required for a smartphone device to run an application depends on various factors, so it’s not possible to determine them with accuracy regarding the Physiomaster app that was analysed. However, all current smartphones have accelerometers embedded in their hardware, for several different functions. Hence, the hardware requirement is fulfilled. With respects to software, the app stores for both operating systems (Android and iOS) have that requirement present when attempting to download the app, which cannot be done if the minimum requirements are not met.
Comment 7: The image that appears does not allow us to see the details; therefore, it is recommended to use one in the foreground.
Response 7: Thank you for your comment. We have added new figures, with increased contrast.
Comment 8: There is no image in the software, and it would be ideal to include an illustrative image.
Response 8: Thank you for your comment. In Figure 2 we have added images of the smartphone software.
Comment 9: Discussion. Data compared with other studies are indicated, but there are no classifications by age or sex, which should appear in the limitations as it does not clarify how to apply the findings.
Response 9: Thank you for your comment. Since the study was conducted on healthy young adults from both genders, its results only pertain to this specific age group. Such limitation was mentioned on the discussion section. We also reiterate that futures studies should address other (older) age groups, to be more realistic of the clinical environment, since they are usually associated with an increase in musculoskeletal conditions.
This manuscript is a resubmission of an earlier submission. The following is a list of the peer review reports and author responses from that submission.
Round 1
Reviewer 1 Report
Comments and Suggestions for Authors
Based on the review of the manuscript, I would recommend the following points to improve its overall clarity and structure:
- Structural Organization:
- The current structure of the manuscript is confusing. After the introduction, the text moves directly to the results and discussion without adequately addressing materials and methods. To ensure logical flow, the methodology should be presented immediately after the introduction, as it provides essential context for understanding the results.
- Figures:
- Figures 1 and 2 are of poor quality and need improvement to enhance readability and understanding. Replacing them with higher-resolution images or clearer graphical representations would be beneficial.
- Table Formatting and Clarity:
- Table 2 lacks a footer or clarification of what the data represent (e.g., cm, degrees). Adding these details would improve the table's readability and comprehension.
- Tables 3 and 4 could be merged to streamline the text and make it more visually appealing. This would reduce redundancy and make the results section more concise.
- Purpose Statement:
- The purpose of the study is unnecessarily repeated at the end of the introduction and the beginning of the discussion. Retaining it at the end of the introduction would suffice.
- Sample Description:
- The sample details are presented twice, first at the beginning of the results section and again in the materials and methods section (divided into participants and sample size calculation). Consolidating this information into one section under materials and methods would improve clarity and avoid redundancy.
- Conclusions:
- Adding a subsection on the practical applications of the findings in other areas or contexts would enhance the relevance of the study. This would help readers and practitioners understand how the research outcomes can be applied more broadly.
While the work is original and innovative, substantial modifications are necessary to improve its structure, presentation , and clarity.
Reviewer 2 Report
Comments and Suggestions for Authors
I recommand to reject the manuscript as not fulfilling basic criteria for the publication.
That are :
1. It is not scientific project and evaluation but special device validation
2. Very small number of participants for any evaluation
3. The manuscript is written in disorderly, results and discussion precede material und methods
4. Pictures enclosed are to small for any presentation of the measurement
Reviewer 3 Report
Comments and Suggestions for Authors
Dear Authors,
Thank you for the opportunity to review your manuscript titled “Validity and reliability of a smartphone application for measuring joint range of motion in healthy individuals compared to 2D software: a cross-sectional observational validation study.” Your work addresses an important topic in physiotherapy by evaluating the utility of a smartphone application for joint range of motion measurements, which could potentially enhance accessibility and efficiency in clinical practice.
Strengths of the Study:
- Practical Application: The PhysioMaster app shows promise as a cost-effective and portable tool for clinical use, with findings supporting its validity and reliability.
- Comprehensive Statistical Analysis: The study employs appropriate statistical methods, including ICC, Pearson correlation, and Bland-Altman plots, to assess the application’s accuracy comprehensively.
Areas for Improvement:
- Sample Diversity: The study focuses on healthy young adults, which limits the generalizability of the findings. Including participants from various age groups and with musculoskeletal conditions would provide more clinically relevant insights.
- Technical Limitations of the App: While the manuscript notes compatibility issues with older smartphone models, further elaboration on the technical requirements and potential solutions would enhance its utility for readers.
- Inter-rater Variability: Although the variability between raters is acknowledged, discussing the impact of evaluator experience and standardizing training protocols could strengthen the reliability claims.
Overall, your study offers valuable contributions to the field and lays the groundwork for future research. Addressing the above points would further enhance the robustness and clinical relevance of your findings.
Thank you for your contributions, and I look forward to seeing your study published.
Sincerely,